# VOCs from Exhaled Breath for the Diagnosis of Hepatocellular Carcinoma

**DOI:** 10.3390/diagnostics13020257

**Published:** 2023-01-10

**Authors:** Thanikan Sukaram, Terapap Apiparakoon, Thodsawit Tiyarattanachai, Darlene Ariyaskul, Kittipat Kulkraisri, Sanparith Marukatat, Rungsun Rerknimitr, Roongruedee Chaiteerakij

**Affiliations:** 1Program in Medical Sciences, Faculty of Medicine, Chulalongkorn University, Bangkok 10330, Thailand; 2Division of Gastroenterology, Department of Medicine, Faculty of Medicine, Chulalongkorn University, Bangkok 10330, Thailand; 3Department of Computer Engineering, Faculty of Engineering, Chulalongkorn University, Bangkok 10330, Thailand; 4Faculty of Medicine, Chulalongkorn University, Bangkok 10330, Thailand; 5Image Processing and Understanding Team, Artificial Intelligence Research Group, National Electronics and Computer Technology Center (NECTEC), Pathum Thani 12120, Thailand; 6Center of Excellence for Innovation and Endoscopy in Gastrointestinal Oncology, Division of Gastroenterology, Faculty of Medicine, Chulalongkorn University, Bangkok 10330, Thailand

**Keywords:** volatile organic compounds (VOCs), biomarkers, cancer diagnosis, breath samples, machine learning, Field Asymmetric Ion Mobility Spectrometry (FAIMS)

## Abstract

Background: Volatile organic compound (VOC) profiles as biomarkers for hepatocellular carcinoma (HCC) are understudied. We aimed to identify VOCs from the exhaled breath for HCC diagnosis and compare the performance of VOCs to alpha-fetoprotein (AFP). The performance of VOCs for predicting treatment response and the association between VOCs level and survival of HCC patients were also determined. Methods: VOCs from 124 HCC patients and 219 controls were identified using the XGBoost algorithm. ROC analysis was used to determine VOCs performance in differentiating HCC patients from controls and in discriminating treatment responders from non-responders. The association between VOCs and the survival of HCC patients was analyzed using Cox proportional hazard analysis. Results: The combination of 9 VOCs yielded 70.0% sensitivity, 88.6% specificity, and 75.0% accuracy for HCC diagnosis. When differentiating early HCC from cirrhotic patients, acetone dimer had a significantly higher AUC than AFP, i.e., 0.775 vs. 0.714, respectively, *p* = 0.001. Acetone dimer classified HCC patients into treatment responders and non-responders, with 95.7% sensitivity, 73.3% specificity, and 86.8% accuracy. Isopropyl alcohol was independently associated with the survival of HCC patients, with an adjusted hazard ratio of 7.23 (95%CI: 1.36–38.54), *p* = 0.020. Conclusions: Analysis of VOCs is a feasible noninvasive test for diagnosing and monitoring HCC treatment response.

## 1. Introduction

Hepatocellular carcinoma (HCC) is one of the leading causes of cancer mortality worldwide [1]. Currently available treatments against HCC yield unsatisfactory outcomes due to the high recurrence rate and heterogenous nature of HCC [2]. Alpha-fetoprotein (AFP) is the most widely used biomarker for HCC. However, AFP at a cutoff of 20 ng/mL provided 52% sensitivity for detecting HCC at any stage and 44% sensitivity for detecting early-stage HCC [3]. AFP had poor sensitivity for monitoring treatment response, as approximately 60% of HCC patients did not have significant changes in AFP levels after treatment [4]. Abdominal ultrasonography is recommended as a screening tool for HCC in at-risk individuals. However, the sensitivity of ultrasonography for the detection of early-stage HCC is limited [5,6]. Although Computerized Tomography (CT) and Magnetic Resonance Imaging (MRI) are the primary radiologic method for diagnosing and evaluating responses to HCC therapy, these modalities are expensive and may have adverse effects [7]. Thus, novel approaches for early detection, diagnosis, and monitoring of therapeutic responses are crucial.

Analysis of volatile organic compounds (VOCs) has increasingly been investigated for their potential as diagnostic biomarkers for a number of cancers, e.g., liver, lung, thyroid, breast, gastrointestinal, gynecological, and urological cancers [8,9,10]. VOCs are compounds produced from tumor cell metabolic activity or the body’s immune response [11]. VOCs profile represents several biological pathways related to cancer development and proliferation, such as inflammatory pathway, oxidative stress pathway, and apoptosis, and reflects abnormal intracellular metabolism related to cancer cell activities, such as altered glucose metabolism and redox dysregulation [12]. One tumor can express a variety of VOCs, which are released into the blood circulation and can be detected in several bodily fluids such as blood, exhaled breath, urine, and feces [11].

The combination of VOCs may address the disadvantages of conventional biomarkers [6,10,13]. Due to tumor heterogeneity, the combination of VOCs could potentially serve as more accurate diagnostic biomarkers than a conventional single biomarker [13]. VOCs can also be used as markers for monitoring treatment responses, for example, in lung cancer patients, with an accuracy of 85% [14]. These signified the role of VOCs as tumor biomarkers for diagnosis and treatment monitoring. Some VOCs can be considered as generic cancer biomarkers due to their associations with several cancers [11]. For example, acetone and hexanal were found in at least eight cancers [11]. These findings suggested that different cancers may express the same VOCs, contributing to the low biomarker specificity. Therefore, research focusing on identifying VOC profiles specific to each cancer is needed.

Previous studies have shown that VOCs can be used as biomarkers for tumors with high mutational heterogeneity, such as HCC [15,16,17,18,19]. The feasibility of using exhaled VOC for detecting HCC was demonstrated in a study of scent detection by a dog reporting a sensitivity of 78% to differentiate HCC patients from healthy individuals [15]. A study enrolling 22 cirrhotic patients reported that patients with HCC had a higher level of acetone but lower levels of isoprene and pentane in exhaled breath than those without HCC [16]. Another research reported that levels of three VOCs, i.e., 3-hydroxy-2-butanone, styrene, and decane, from exhaled breath of HCC patients were significantly higher than the breath of healthy volunteers [17]. A very recent study showed that a combination of 18 exhaled VOCs had an overall accuracy of 72% in differentiating HCC patients from non-HCC subjects [18]. Although these studies demonstrated the potential of VOCs as biomarkers for HCC diagnosis, the performance of VOCs compared to AFP has yet to be investigated, and whether VOCs can be used to monitor treatment response and determine the prognosis of HCC patients remains unknown.

Several techniques can be applied to profiling VOCs. Gas chromatography-mass spectrometry (GC-MS) and selective ion flow tube mass spectrometry (SIFT-MS) are commonly used due to their high resolution and sensitivity. However, they are costly, time-consuming, and require expertise for operation. An electronic nose (E-nose) sensor, a small-size portable analyzer, is more user-friendly and yields rapid results at a lower cost. It is therefore proposed to be used as a point-of-care testing analyzer [20]. Nonetheless, this type of sensor is sensitive to water vapor and has a relatively short lifespan. Another technique is Field Asymmetric Ion Mobility Spectrometry (FAIMS), which provides comparable sensitivity to GC-MS and SIFT-MS but is less expensive and less sophisticated, thus, more suitable for VOC analysis for clinical use [21].

Artificial intelligence (AI) has been increasingly developed and utilized in healthcare to assist medical personnel, such as computational pathology and chest x-ray screening [22]. Regarding VOC analysis, AI was applied to select the ideal combination of VOCs for diagnosing cancerous and non-cancerous diseases, e.g., tuberculosis, lung cancer, and HCC [19,23,24,25]. Given the complexity and heterogeneity of tumor biology, the prediction model constructed using AI rather than conventional analyses is more capable of identifying the VOC pattern that differentiates cancer patients from individuals without cancer [26].

In this study, we aimed to (1) identify the potential VOCs as biomarkers for HCC diagnosis using the AI algorithm, (2) investigate the diagnostic performance of VOCs in comparison to AFP, (3) assess the feasibility of utilizing VOCs for HCC treatment monitoring, and (4) determine an association between the survival of HCC patients and VOCs profile to explore the potential role of VOCs as prognostic biomarkers.

## 2. Materials and Methods

### 2.1. Participants

VOCs as biomarkers for HCC diagnosis. We recruited 124 HCC patients, 124 cirrhosis patients, and 95 healthy volunteers from the Center of Excellence for Innovation and Endoscopy in Gastrointestinal Oncology, Division of Gastroenterology, Department of Medicine, Chulalongkorn University. For HCC patients, the inclusion criteria were patients newly diagnosed with HCC by histopathologic confirmation or typical radiologic characteristics according to the American Association for the Study of Liver Diseases criteria [27] and who had not received any oncologic treatment. HCC stage was classified according to the Barcelona-Clinic Liver Cancer (BCLC) staging system [28]. This study classified HCC stages as early HCC (BCLC stages 0 and A) and advanced HCC (BCLC stages B and C). The exclusion criteria were patients with recurrent HCC or a history of other cancers. Cirrhosis was diagnosed by histopathology and/or radiologic features of small-sized nodular surface liver and evidence of portal hypertension, e.g., intraabdominal collateral circulation and/or splenomegaly. The severity of cirrhosis was classified by Child-Turcotte-Pugh (CTP) class. Healthy volunteers were participants who had normal liver function tests and did not have a history of chronic liver diseases and malignancies. The pre-treatment breath samples were collected from all HCC patients prior to any oncogenic treatment.

VOCs as biomarkers for monitoring treatment response. A subgroup of 38 patients was randomly selected from a cohort of 124 HCC patients. This subgroup was treated with either transarterial chemoembolization (TACE) or percutaneous local ablative therapy (PLAT) with radiofrequency or microwave ablation. Patients were followed up at 1 month post-treatment to track for the change in VOCs profile. On the follow-up day, CT or MRI imaging was performed, and the post-treatment breath samples were collected. Treatment response was determined according to the modified Response Evaluation Criteria in Solid Tumors (mRECIST) criteria. Patients without remaining viable tumors were considered as a treatment response group, whereas those with remaining viable tumors were a non-response group.

### 2.2. Clinical Data Collection

Participants’ baseline characteristics including gender, age, smoking status, alcohol intake, cirrhosis, and etiologies of chronic liver disease (chronic viral hepatitis B and C infection, non-alcoholic fatty liver disease (NAFLD), alcoholic liver disease, or others) and diabetes mellitus were obtained from electronic medical records. We also abstracted information regarding HCC BCLC stages, liver function tests, and AFP levels. For assessment of survival, HCC patients were followed up for their vital status until death or the end of the study period (22 March 2022).

### 2.3. Breath Collection

Prior to breath sampling, all participants had been fasting, ceased smoking and exercising, and stopped current medication for at least 6 h to lessen exogenous confounders that could contaminate the breath. A total of 100 mL end-exhaled-air was collected directly from participants’ noses via mask at a rate of 50 mL/min using the ReCIVA^®^ breath sample system (Owlstone Medical, Cambridge, UK). The ReCIVA^®^ breath sample system was connected to a computer, and real-time breath monitoring was displayed via ReCIVA^®^ software. This system provided a constant supply of pure oxygen to the patient’s mask. Finally, the exhaled breath samples were concentrated in 4 thermal desorption (TD) tubes (Unity™, Markes International Ltd., Llantrisant, UK). The samples were further analyzed by GC-FAIMS immediately after breath collection (Figure 1).

### 2.4. VOCs Extraction and Measurement

VOCs were extracted from TD tubes with a constant flow of helium at 50 mL/min using 2 conditions. In the first condition, the TD tube was dry purged for 1 min and heated at 280 °C for 5 min. In the second condition, the cold trap (U-T12ME-2S, Markes International Ltd., Llantrisant, UK) was temperature programmed from 10 °C to 290 °C.

Extracted VOCs were transferred to GC by capillary line heated at 130 °C. The GC system (Thermo Scientific TRACE1310 GC, Waltham, MA, USA) used HP-PLOT U GC column (30 m × 0.32 mm ID × 10 µm df) (Agilent Technologies, Santa Clara, CA, USA) and helium as the carrier gas with a flow rate of 1.0 mL/min for VOCs separation. The GC column was initially heated at 40 °C for 2 min and ramped to 130 °C with a rate of 10 °C/min.

Separated VOCs were sent to FAIMS via a drift tube with a length of 7.5 cm and a drift voltage of 5 kV by a transfer line heated at 130 °C for VOC detection. The FAIMS system (Owlstone Medical Lonestar VOC Analyzer FAIMS system, Cambridge, UK) was operated at a temperature of 40 °C and an ambient pressure of 10 millibars. Purified air was used as drift gas with a flow rate of 150 mL/min. The VOCs profile was shown as a chromatogram. VOC identification was achieved by calibrating the retention time of standard solutions with the chromatogram. System calibration was performed daily to ensure the accuracy of all instruments.

### 2.5. Data Analysis

For baseline characteristics, continuous variables were reported as mean ± standard deviation (SD) or median and range as appropriate and compared by one-way ANOVA test. Categorical variables were reported as numbers (%) and compared by Pearson’s Chi-square test.

VOC levels were expressed as median (range) arbitrary units (AU). The VOCs levels were compared among HCC, cirrhosis, and healthy groups using one-way ANOVA and between HCC and cirrhosis groups using the Mann–Whitney U test. Logistic regression analysis was performed to evaluate the association between VOCs and HCC.

In this study, the XGBoost was used to identify important features of VOC profiles, and the algorithm also selected the ideal combination of VOCs to yield optimal statistical parameters. XGBoost algorithm is one of the AI algorithms. We opted to use the XGBoost algorithm as it minimized false predictions by continually learning from the model previously constructed as a series and therefore maximized the efficiency of the prediction model [29]. The algorithm rated the importance of each VOC feature by counting the frequency that the VOC feature was able to differentiate among the three cohorts and reported the rating as a feature importance score (F score). The participant cohort was split into 2 datasets, i.e., 80% as a training set and 20% as a test set. The training set comprised 101 HCC patients, 100 cirrhotic patients, and 75 healthy volunteers, while the test set included 23 HCC patients, 24 cirrhotic patients, and 20 healthy volunteers. The model generated from the training set was evaluated for its predictive performance using the test set. The model performance was reported as sensitivity, specificity, positive predictive value (PPV), negative predictive value (NPV), and accuracy. Furthermore, XGBoost was used to construct the model for differentiating early HCC and advanced HCC.

Receiver-operating characteristic (ROC) curve was generated to identify the optimal cutoff of VOCs and AFP for HCC diagnosis. Area Under the ROC Curve (AUC) of AFP levels at the cutoff identified by the ROC curve, the median AFP value, and the AFP of 20 ng/mL were estimated. The level of AFP that yielded the maximum AUC was selected as the optimal cutoff for AFP. Concordance statistics (c-statistics) were used to compare the diagnostic performance of VOCs and AFP.

A paired sample t-test and Mann–Whitney U test were performed to assess the change in median levels of VOCs in HCC patients between the pre-and post-treatment groups and between the treatment response and non-response groups, respectively. ROC analysis was performed to determine the performance of VOCs as biomarkers for monitoring treatment response.

The correlation between VOCs and the survival of HCC patients was evaluated using Cox proportional hazard analysis. Age, gender, and other factors related to patient survival were also analyzed by univariate and multivariate Cox proportional hazard model.

Statistical analyses were performed using the SPSS package version 22.0.0 (SPSS Inc., Chicago, IL, USA). A *p*-value of < 0.05 was considered statically significant.

## 3. Results

### 3.1. Baseline Characteristics

The baseline characteristics of the study cohort are shown in Table 1. The number of HCC patients with BCLC stages 0, A, B, and C were 25 (20.2%), 42 (33.9%), 37 (29.8%), and 20 (16.1%), respectively. Age, gender, smoking status, and alcohol consumption were not statistically different among the HCC, cirrhosis, and healthy groups. There was no significant difference in etiologies of chronic liver disease between HCC and cirrhosis groups. The proportion of patients with CTP class A, B, and C cirrhosis differed significantly between the HCC (78.2%, 21.0%, and 0.8%) and cirrhosis (93.5%, 6.5%, and 0.0%) groups, *p* = 0.002. The HCC group had significantly higher levels of total bilirubin, aspartate aminotransferase, alanine aminotransferase, alkaline phosphatase, and AFP but a lower level of albumin than the cirrhosis and healthy groups.

### 3.2. VOCs as Biomarkers for HCC Diagnosis

The XGBoost algorithm identified nine VOCs as important features for differentiating among HCC, cirrhosis, and healthy groups (Figure 2A). There were four VOCs that had noticeably high F scores, i.e., acetone monomer (332 scores), followed by ethanol (286 scores), acetone dimer (280 scores), and acetonitrile (268 scores).

Figure 2B shows the results of the predictive model generated by the XGBoost algorithm in classifying HCC, cirrhosis, and healthy controls. The model’s overall accuracy in correctly classifying the participants into three groups was 75.0%, with 70.0% sensitivity, 88.6% specificity, 76.2% PPV, 84.8% NPV, and 82.1% accuracy in differentiating HCC from cirrhosis and healthy groups. When differentiating cirrhosis from the other two groups, the sensitivity, specificity, PPV, NPV, and accuracy were 79.2%, 81.4%, 70.4%, 87.5%, and 80.6%, respectively. Lastly, healthy controls were correctly identified with a sensitivity of 75.0%, specificity of 91.5%, PPV of 78.9%, NPV of 89.6%, and accuracy of 86.5%.

Table 2 shows the levels of nine VOCs identified by the XGBoost algorithm. There were significant differences in the levels of acetone monomer (*p* < 0.001), 1,4-petadiene (*p* = 0.015), isopropyl alcohol (*p* = 0.022), acetone dimer (*p* < 0.001), and toluene (*p* < 0.001) among the HCC, cirrhosis, and healthy groups. In HCC patients, the acetone monomer level was lower, while the levels of acetone dimer and isopropyl alcohol were higher than in cirrhosis and healthy groups. The HCC group had higher levels of 1,4-pentadiene and toluene than the healthy group but lower levels of these two VOCs than the cirrhosis group. When compared between HCC and cirrhosis groups, the HCC group significantly had higher levels of isopropyl alcohol and acetone dimer than the cirrhosis group, with median (range) levels of 0.18 (0.07–1.73) vs. 0.15 (0.08–1.22) AU, and 5.08 (2.84–5.83) vs. 4.48 (2.59–4.93) AU for isopropyl alcohol and acetone dimer, respectively, *p* = 0.032 and *p* < 0.001.

Regarding the differentiation between the stages of HCC, five VOCs (ethanol, acetone dimer, benzene, 1,4-pentadiene, and isopropyl alcohol) were identified as important features for classifying patients with early and advanced HCC (Figure 2C). The model correctly predicted 11 of 14 early HCC patients and 8 of 9 advanced HCC patients, accounting for 78.6% sensitivity, 88.9% specificity, 72.7% PPV, 91.7% NPV, and 82.6% accuracy (Figure 2D).

### 3.3. Performance of Acetone and AFP as Diagnostic Biomarkers

Acetone dimer was the VOC that significantly differed when comparing HCC to the cirrhosis group (*p* < 0.001) and between the three groups (*p* < 0.001). We performed logistic regression analysis to determine whether acetone dimer was independently associated with HCC among cirrhotic patients. By multivariate analysis adjusted for age, gender, and CTP class, we found that acetone dimer and AFP were significantly associated with HCC, with adjusted odd ratios (OR) of 9.29 (95%CI: 2.34–36.79) and 1.42 (95%CI: 1.06–1.88), respectively, *p* < 0.001 and 0.006. Thus, acetone dimer was selected as a candidate VOCs for the diagnosis of HCC. For the diagnostic performance of acetone dimer at a cutoff level of 4.6579 AU, the ROC curve differentiated between HCC patients and non-HCC patients (combined cirrhosis and healthy patients), with an AUC of 0.816 and 83.9% sensitivity, 79.4% specificity, 70.0% PPV, 89.6% NPV, and 81.0% accuracy (Table 3). At the same cutoff level, HCC patients were differentiated from cirrhotic patients, with an AUC of 0.769 and 83.9% sensitivity, 69.9% specificity, 73.8% PPV, 81.1% NPV, and 76.9% accuracy. When distinguished between the early HCC and cirrhosis groups, the AUC was 0.775, with 85.1% sensitivity, 69.9% specificity, 60.6% PPV, 89.6% NPV, and 75.2% accuracy.

Regarding the AFP as the biomarker for HCC diagnosis, the AFP level of 20 ng/mL provided the greatest performance among the three pre-specified cutoffs (data not shown). AFP at a cutoff of 20 ng/mL differentiated between HCC and non-HCC groups, with AUC of 0.802 and 62.4% sensitivity, 100% specificity, 100% PPV, 62.4% NPV, and 69.9% accuracy (Table 3). The HCC group was differentiated from the cirrhosis group, with an AUC of 0.806 and 40.2% sensitivity, 100% specificity, 100% PPV, 58.8% NPV, and 67.7% accuracy.

For the comparison between the performance of acetone dimer and AFP as a biomarker for HCC diagnosis, acetone dimer significantly had higher AUCs than AFP for diagnosis of HCC at any stage and HCC at an early stage, i.e., 0.816 vs. 0.802 and 0.775 vs. 0.714, respectively, *p* = 0.002 and 0.001, indicating that acetone dimer had a better overall performance than AFP for HCC diagnosis.

### 3.4. VOCs as Biomarkers for Monitoring Treatment Response

Of the 38 HCC patients of whom pre- and post-treatment exhaled VOCs were profiled, 22 patients were treated with TACE, and 16 patients underwent PLAT with radiofrequency or microwave ablation. After the treatment, there were significant reductions in the median (range) levels of 4 VOCs as follows: dimethyl sulfide from 0.67 (0.17–1.39) to 0.48 (0.15–1.38) AU, *p* = 0.007, benzene from 0.15 (0.06–0.72) to 0.11 (0.03–0.45) AU, *p* = 0.013, acetone dimer from 5.10 (2.98–5.83) to 4.35 (2.70–5.33) AU, *p* < 0.001, and acetonitrile from 0.15 (0.10–3.67) to 0.10 (0.06–0.40) AU, *p* = 0.006 (Appendix A). At follow-up, 23 (60.5%) patients (12 TACE, 11 PLAT) responded to treatment, while the other 15 (39.5%) patients (10 TACE, 5 PLAT) did not respond. The magnitudes of dimethyl sulfide and acetone dimer reductions in the treatment response group were significantly greater than in the non-response group. (Appendix A). The reduction magnitude in the response group and non-response group was −0.15 (−0.71, 0.32) vs. 0.05 (−0.28, 0.32) AU for dimethyl sulfide, and −1.32 (−2.75, −0.06) vs. −0.12 (−2.63, 1.80) AU for acetone dimer, respectively, *p* = 0.031 and 0.003.

The performance of acetone dimer in differentiating HCC patients who responded to treatment from the non-responders was determined. At the cutoff of 0.2591 AU, acetone dimer provided an AUC of 0.780, with 95.7% sensitivity, 73.3% specificity, 84.6% PPV, 91.7% NPV, and 86.8% accuracy.

### 3.5. Association between VOCs and Survival of HCC Patients

The median follow-up time for HCC patients was 7 months. There were 90.3% (*n* = 112/124) of HCC patients who survived until the end of the study. We found that advanced HCC stage, high levels of AFP, and isopropyl alcohol were significantly associated with worse survival, with a hazard ratio (HR) of 7.07 (95%CI: 1.55–32.31), 1.53 (95%CI: 1.09–2.17) and 4.84 (95%CI: 1.28–18.24), respectively, *p* = 0.012, 0.015 and 0.020 (Table 4). By multivariate analysis, isopropyl alcohol remained independently associated with worse survival, with an adjusted HR of 7.23 (95%CI: 1.36–38.54), *p* = 0.020; and advanced HCC stage was borderline significantly associated with the decreased survival, with an adjusted HR of 5.66 (95%CI: 0.90–35.59), *p* = 0.065 (Table 4).

## 4. Discussion

In this study, we used the XGBoost algorithm to identify the unique features of the exhaled VOCs and generate the predictive model for HCC diagnosis. The model classified patients into the correct groups with satisfactory performance, with an overall accuracy of 75%.

The performance of exhaled VOCs for HCC diagnosis observed in this study (70.0% sensitivity, 88.6% specificity) was consistent with those shown in previous studies, with reported sensitivities ranging from 73.0% to 86.7% and specificities ranging from 71.0% to 91.7% [15,17,18,19]. Likewise, the performance of VOCs in differentiating early HCC from advanced HCC (78.6% sensitivity, 88.9% specificity) was comparable to the previously reported sensitivity of 76.5% and specificity of 82.7% [19]. These findings support the utility of VOC analysis for HCC diagnosis. The VOCs profile could possibly assist physicians in determining the HCC stages and executing an appropriate treatment plan, as the standard treatment modality differed between early and advanced HCC.

Among the five VOCs that distinguished HCC patients from those without HCC, the levels of the two VOCs, acetone dimer and isopropyl alcohol, were significantly greater in HCC patients than in non-HCC patients. These observations were in line with previous studies [18,19] A study including 112 HCC patients, 30 cirrhotic patients, and 54 participants without chronic liver diseases reported that the level of acetone in the exhaled breath was found to be significantly higher in HCC patients and cirrhotic patients than participants without chronic liver diseases. Moreover, a combination of 18 exhaled VOCs, which included acetone and isopropyl alcohol, provided 73% sensitivity and 71% specificity for discriminating HCC patients from non-HCC individuals without cirrhosis [18]. A very recent study of 97 HCC patients and 111 non-HCC patients reported that HCC patients had a significantly higher level of acetone than non-HCC patients. When acetone was combined with another five exhaled VOCs, the combination of VOCs showed 76.5% sensitivity and 82.7% specificity for HCC diagnosis. Similar to the previous study, this study reported a significant reduction in acetone levels after HCC therapy, with a greater reduction in patients who responded to therapy when compared to those who did not respond to the treatment [19]. Taken together, these findings highlight the potential role of acetone as a biomarker for HCC diagnosis and monitoring treatment response.

Acetone is a ketone compound derived from the spontaneous degradation of acetoacetate, a ketone body generated from acetyl-CoA, mainly obtained from the beta-oxidation of long-chain fatty acids [30]. Another source of acetone is the alcohol dehydrogenase enzyme, which reversibly converts isopropyl alcohol to acetone via an oxidation-reduction reaction [31]. This interaction implies that acetone is directly correlated to isopropyl alcohol and these two VOCs usually co-exist [11,31,32]. These two VOCs are associated with cellular metabolic dysregulation. The reprogramming of energy metabolism is one of the pathogeneses of HCC [33]. Cancer cells abnormally metabolize glucose and deplete the availability of glucose for the surrounding cells, hence shifting the energy source toward the lipid metabolism pathway [34]. Acetone is one of the byproducts of the fatty acid oxidation process, which is also upregulated to produce sufficient energy for cancer growth and the anabolic process [35]. For these reasons, metabolic dysregulation was responsible for the rise of these two VOCs in HCC patients.

Although accumulating evidence showed the potential of acetone as biomarkers for HCC diagnosis, the usage of acetone and isopropyl alcohol in HCC was scarcely investigated; this study further examined the clinical applications for these VOCs. Firstly, we found that HCC patients significantly expressed a decreased level of acetone dimer after receiving cancer therapy. Moreover, the patients who responded to treatments also expressed a lower level of acetone dimer in comparison to those in the non-response group. Acetone dimer provided high sensitivity, specificity, and accuracy for predicting the treatment responsiveness in HCC patients. These findings implied that the changes in acetone level had the potential role as biomarkers for monitoring treatment responses.

We found that acetone dimer showed a significantly greater AUC than AFP, indicating better overall performance for HCC diagnosis. Given the limited performance of AFP, exhaled VOCs may be useful as an additional tool to enhance the investigations of HCC detection.

The present study found that an increased level of isopropyl alcohol was significantly associated with a decline in survival for HCC patients. The results implicated that isopropyl alcohol may be used as HCC prognostic biomarker. In clinical settings, prognostic biomarkers might be useful in guiding appropriate treatment plans and physicians’ decisions. Thus, HCC patients with elevated isopropyl alcohol were considered at risk and should be closely monitored for disease progression.

Unlike most studies of exhaled VOCs for diagnosis of HCC, to the best of our knowledge, this is the first study demonstrating that the exhaled VOC yielded better overall performance than AFP for diagnosis of HCC at any stage and HCC at an early stage. The number of HCC patients and controls was relatively large compared to previous studies. Another strength of this study was that we applied FAIMS with the XGBoost algorithm for VOC profile analysis. In most previous studies, VOCs were identified using GC-MS, the complicated technique that is difficult to apply for large-scale routine clinical service [36,37]. Herein, our study demonstrated the feasibility of a new technique, GC-FAIMS with ReCIVA^®^ breath sample system, for VOC analysis in a point-of-care setting. Accordingly, exhaled VOCs might be used as an adjunctive screening tool to overcome the limitations of ultrasound machines, such as undetectable small tumors and being operator-dependent [38]. Taken together, the detection of exhaled VOCs could offer a novel approach that might lessen HCC diagnostic burdens in real clinical practice.

There are some limitations to our study. We did not compare the performance of exhaled VOCs with ultrasound, the standard tool for HCC screening. Although our participants had various risk factors representing common contributing factors associated with HCC development, most of HCCs occurred in patients with underlying chronic viral hepatitis B infection. Because HCCs occurring from different etiologies have different genetic alterations and dysregulated pathways, it remains unknown whether the VOCs identified in this study could be used as diagnostic biomarkers in another population of whom chronic viral hepatitis C infection and alcoholic cirrhosis are the main risk factors for HCC. The predictive model in this study was derived from a single cohort, and the model performance was evaluated in a cohort of participants from the same center. The generalizability of this model has to be further investigated. External validation of these findings in another independent cohort is required before applying this approach to clinical practice. Exogenous confounding factors such as environmental factors, occupational exposure, and underlying diseases could have possibly been involved in the production of VOCs and altered the VOCs profiles. Therefore, we implemented the breath collection protocol to minimize these confounding factors. Analysis of VOCs using the TD-GC-FAIMS technique requires the creation of a VOC library. The machine is relatively large, thus, impeding its use as a screening tool in field conditions. Optimizing the analysis by a targeted approach and further developing a portable analyzer with sensors to detect some selected VOCs would facilitate the utility of exhaled VOC analysis for cancer diagnosis in clinical practice.

## 5. Conclusions

Exhaled VOCs provide promising performance as non-invasive biomarkers for HCC diagnosis, staging, and monitoring of HCC treatment response. Acetone has a better overall performance than the standard biomarker AFP for HCC diagnosis, and isopropyl alcohol is a predictor of survival in HCC patients. Validating these findings in other cohorts of individuals at-risk for HCC is warranted.

## Figures and Tables

**Figure 1 diagnostics-13-00257-f001:**
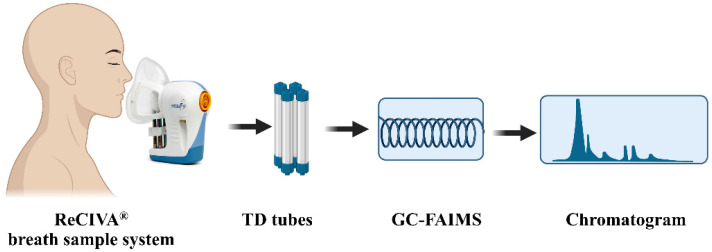
Schematic diagram of breath sample collection and breath sample analysis.

**Figure 2 diagnostics-13-00257-f002:**
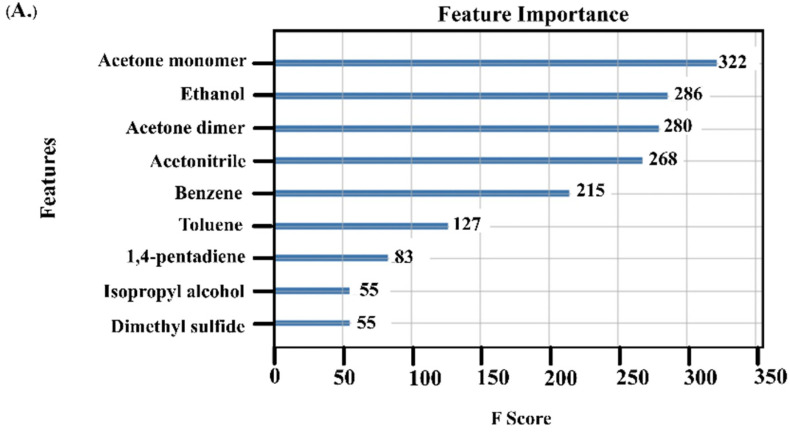
(**A**) A feature importance score (F score) of VOCs used to discriminate among HCC, cirrhosis, and healthy controls, (**B**) The confusion matrix from XGBoost to classify participants as HCC, cirrhosis, and healthy controls, (**C**) F score of VOCs to discriminate between the early (BCLC stages 0-A) and advanced (BCLC stages B-C) HCC stages, (**D**) The confusion matrix from XGBoost to differentiate between early HCC and advanced HCC.

**Table 1 diagnostics-13-00257-t001:** Baseline characteristics of the study cohort.

Variables	HCC (*n* = 124)	Cirrhosis (*n* = 124)	*p* ^†^	Healthy (*n* = 95)	*p* ^‡^
Age, years *	62.7 ± 12.6	60.6 ± 9.2	0.126	59.3 ± 9.1	0.053
Male, n (%)	60 (48.4%)	62 (50.0%)	0.799	47 (49.5%)	0.967
Smoking, n (%)	58 (46.8%)	48 (38.7%)	0.199	35 (36.8%)	0.265
Alcohol consumption, n (%)	23 (18.5%)	27 (21.8%)	0.527	22 (23.2%)	0.684
Cirrhosis, n (%)	122 (98.4%)	124 (100.0%)	0.156	0 (0.00%)	0.016
Child-Pugh class, n (%)			0.002		
A	97 (78.2%)	115 (93.5%)		0 (0.0%)	
B	26 (21.0%)	8 (6.5%)		0 (0.0%)	
C	1 (0.8%)	0 (0.0%)		0 (0.0%)	
Chronic HBV infection, n (%)	53 (42.7%)	42 (33.9%)	0.151	0 (0.0%)	
Chronic HCV infection, n (%)	34 (27.4%)	48 (38.7%)	0.059	0 (0.0%)	
NAFLD, n (%)	37 (29.8%)	32 (25.8%)	0.479	0 (0.0%)	
Diabetes mellitus, n (%)	45 (36.3%)	41 (33.1%)	0.594	0 (0.0%)	
Albumin (g/dL) *	3.8 ± 0.6	4.0 ± 0.4	0.001	4.4 ± 0.2	<0.001
Total bilirubin (mg/dL) *	1.6 ± 2.2	1.1 ± 0.9	0.018	0.8 ± 0.3	0.021
AST (U/L) *	74 ± 84	38 ± 36	<0.001	23 ± 6	<0.001
ALT (U/L) *	53 ±78	33 ± 43	0.014	24 ± 12	0.016
Alkaline phosphatase (U/L) *	164 ± 225	94 ± 48	0.001	69 ± 21	0.001
AFP (ng/mL), median (range)	9.9 (1.1–158,906.2)	2.4 (1.0–18.1)	0.008	3.1 (1.3–10.6)	<0.001

* Data are shown as mean ± standard deviation. *p*
^†^ for HCC vs. Cirrhosis, *p*
^‡^ for HCC vs. Cirrhosis vs. Healthy. Abbreviations: AFP—alpha fetoprotein, ALT—alanine aminotransferase, AST—aspartate aminotransferase, HBV—hepatitis B virus, HCV—hepatitis C virus, NAFLD—non-alcoholic fatty liver disease.

**Table 2 diagnostics-13-00257-t002:** VOC levels in HCC, cirrhosis, and healthy groups *.

VOCs	HCC (*n* = 124)	Cirrhosis (*n* = 124)	*p* ^†^	Healthy (*n* = 95)	*p* ^‡^
Ethanol	0.25 (0.12–1.01)	0.25 (0.12–0.44)	0.728	0.26 (0.12–0.42)	0.341
Acetone monomer	3.85 (2.20–4.76)	3.90 (2.28–4.82)	0.559	4.29 (2.07–4.70)	<0.001
Dimethyl sulfide	0.29 (0.04–3.67)	0.41 (0.08–1.81)	0.939	0.28 (0.10–1.24)	0.845
1,4-pentadiene	0.74 (0.00–2.12)	0.82 (0.01–3.35)	0.389	0.65 (0.10–1.54)	0.015
Benzene	0.21 (0.00–0.74)	0.21 (0.01–1.11)	0.199	0.18 (0.10–0.79)	0.139
Isopropyl alcohol	0.18 (0.07–1.73)	0.15 (0.08–1.22)	0.032	0.15 (0.08–1.20)	0.022
Acetone dimer	5.08 (2.84–5.83)	4.48 (2.59–4.93)	<0.001	3.90 (1.32–5.03)	<0.001
Acetonitrile	0.16 (0.07–0.69)	0.15 (0.09–0.70)	0.426	0.17 (0.05–0.74)	0.073
Toluene	0.41 (0.14–2.33)	0.43 (0.09–2.75)	0.782	0.31 (0.06–2.87)	<0.001

* Data are shown as median (range) in arbitrary units. *p*
^†^ for HCC vs. Cirrhosis, *p*
^‡^ for HCC vs. Cirrhosis vs. Healthy.

**Table 3 diagnostics-13-00257-t003:** Performance of acetone dimer and alpha fetoprotein for diagnosis of HCC.

Biomarker	Group *	Cutoff	AUC	Sensitivity	Specificity	PPV	NPV	Accuracy
Acetone dimer	HCC vs. non-HCC	4.6579	0.816	83.9%	79.4%	70.0%	89.6%	81.0%
	Early HCC vs. non-HCC	4.6579	0.822	85.1%	79.4%	55.9%	94.5%,	80.7%
	HCC vs. cirrhosis	4.6579	0.769	83.9%	69.9%	73.8%	81.1%	76.9%
	Early HCC vs. cirrhosis	4.6579	0.775	85.1%	69.9%	60.6%	89.6%	75.2%
AFP	HCC vs. non-HCC	20	0.802	62.4%	100.0%	100.0%	62.4%	69.9%
	Early HCC vs. non-HCC	20	0.721	19.0%	100.0%	100.0%	69.5%,	68.7%
	HCC vs. cirrhosis	20	0.806	40.2%	100.0%	100.0%	58.8%	67.7%
	Early HCC vs. cirrhosis	20	0.714	19.0%	100.0%	100.0%	66.2%	12.6%

* Number of participants in each group is as follows: 124 HCC, 67 Early HCC, 210 non-HCC, and 124 cirrhosis. Abbreviations: AFP—alpha fetoprotein, HCC—hepatocellular carcinoma, AUC—area under the curve, PPV—positive predictive value, and NPV—negative predictive value.

**Table 4 diagnostics-13-00257-t004:** Factors associated with survival of HCC patients.

Variables	Univariate Analysis	*p*	Multivariate Analysis	*p*
	HR (95%CI)		Adjusted HR (95%CI)	
Age	1.00 (0.95–1.05)	0.956	1.03 (0.97–1.09)	0.408
Sex	1.51 (0.48–4.76)	0.483	1.05 (0.21–5.15)	0.957
AFP	1.53 (1.09–2.17)	0.015	1.39 (0.72–2.66)	0.325
HCC stage				
Early (BCLC 0-A)	reference			
Advanced (BCLC B-C)	7.07 (1.55–32.31)	0.012	5.66 (0.90–35.59)	0.065
Acetone monomer	0.54 (0.22–1.31)	0.170		
Dimethyl sulfide	0.79 (0.17–3.57)	0.759		
Benzene	0.64 (0.01–37.49)	0.827		
Isopropyl alcohol	4.84 (1.28–18.24)	0.020	7.23 (1.36–38.54)	0.020
Acetone dimer	1.50 (0.43–5.26)	0.524		
Toluene	0.64 (0.08–5.18)	0.674		

Abbreviations: AFP—alpha fetoprotein, BCLC stage—Barcelona clinic liver cancer stage, HR—Hazard ratio, 95%CI—95% confidence interval.

## Data Availability

The data are available upon requests.

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
