# Peer review of "VOCs from Exhaled Breath for the Diagnosis of Hepatocellular Carcinoma"

_diagnostics, 2023, doi:10.3390/diagnostics13020257_

Round 1

Reviewer 1 Report

Introduction must be revised by adding few more recent research findings on the topic including HCC and other cancers as well. A comparison has to be made between those findings and this work in terms of overall performance.

The text font must be made uniform throughout the manuscript.

Figure 2C must be corrected, the texts are not readable.

Which is the most important parameter to evaluate the classification results, overall accuracy, sensitivity, specificity, or AUC/ROC? Please explain. 

Please highlight the major limitations and disadvantages of the proposed approach.

Some relative papers may enrich the background of Introduction as references:

DOI: 10.1016/j.bbcan.2021.188644

- https://doi.org/10.3390/diagnostics12020491

DOI: 10.1080/08941939.2019.1586015

- https://doi.org/10.3390/diagnostics12020430

DOI: 10.1093/bjsopen/zrab013

The conclusion must be extended by adding the important results from the study and highlighting the take-home-message and future perspectives from the current work.

English revision and grammatical corrections are required.

Author Response

We thank the reviewer for all suggestions. We have reviewed the comments and have considered them carefully. The point-by-point responses to reviewers are in the attached file.

Reviewer 2 Report

I have found this manuscript well-written and structured. There are some vital points needed to be added or revised before considering it further. Here are the comments needed to be addressed: 

1. The cellular metabolic processes and their changes are manifested in emitted volatile organic compound (VOC) compositions of different diseased cells, such as cancer cells will open up the possibilities of early detection techniques. The INTRODUCTION section needs information and background on metabolic changes and VOCs and their role in the detection of diseases and cancer. Please also highlight in the section, the other techniques such as GC-MS and LC-MS, also cyber detection. Some recent citations needed to be added, such as the following one from a very nice journal, explaining a technique of insect brain-based cancer VOC detection using AI techniques. Breath samples could be used easily with this kind of setup. 

https://doi.org/10.1016/j.bios.2022.114814 

Please also highlight the importance of these techniques for non-invasive early detection.

2. During the clinical data collection, was there any data on oral hygiene status? If so, please include it. 

3. Please write the strengths and limitations of the study. 

Author Response

(The authors gave the same response as above.)
